# Heat Stress-Tolerant Quantitative Trait Loci Identified Using Backcrossed Recombinant Inbred Lines Derived from Intra-Specifically Diverse *Aegilops tauschii* Accessions

**DOI:** 10.3390/plants13030347

**Published:** 2024-01-24

**Authors:** Monir Idres Yahya Ahmed, Nasrein Mohamed Kamal, Yasir Serag Alnor Gorafi, Modather Galal Abdeldaim Abdalla, Izzat Sidahmed Ali Tahir, Hisashi Tsujimoto

**Affiliations:** 1United Graduate School of Agricultural Sciences, Tottori University, Tottori 680-8550, Japan; menoo99999@gmail.com; 2Arid Land Research Center, Tottori University, Tottori 680-0001, Japan; renokamal@tottori-u.ac.jp (N.M.K.); izzatahir@yahoo.com (I.S.A.T.); 3Agricultural Research Corporation (ARC), Wad-Medani P.O. Box 126, Sudan; yasirserag@tottori-u.ac.jp (Y.S.A.G.); modatherarc@yahoo.com (M.G.A.A.); 4International Platform for Dryland Research and Education, Tottori University, Tottori 680-0001, Japan

**Keywords:** wheat, heat stress, quantitative trait locus (QTL), GRAS-Di, backcrossed recombinant inbred line (BIL), *Aegilops tauschii*

## Abstract

In the face of climate change, bringing more useful alleles and genes from wild relatives of wheat is crucial to develop climate-resilient varieties. We used two populations of backcrossed recombinant inbred lines (BIL1 and BIL2), developed by crossing and backcrossing two intra-specifically diverse *Aegilops tauschii* accessions from lineage 1 and lineage 2, respectively, with the common wheat cultivar ‘Norin 61′. This study aimed to identify quantitative trait loci (QTLs) associated with heat stress (HS) tolerance. The two BILs were evaluated under heat stress environments in Sudan for phenology, plant height (PH), grain yield (GY), biomass (BIO), harvest index (HI), and thousand-kernel weight (TKW). Grain yield was significantly correlated with BIO and TKW under HS; therefore, the stress tolerance index (STI) was calculated for these traits as well as for GY. A total of 16 heat-tolerant lines were identified based on GY and STI-GY. The QTL analysis performed using inclusive composite interval mapping identified a total of 40 QTLs in BIL1 and 153 QTLs in BIL2 across all environments. We detected 39 QTLs associated with GY-STI, BIO-STI, and TKW-STI in both populations (14 in BIL1 and 25 in BIL2). The QTLs associated with STI were detected on chromosomes 1A, 3A, 5A, 2B, 4B, and all the D-subgenomes. We found that QTLs were detected only under HS for GY on chromosome 5A, TKW on 3B and 5B, PH on 3B and 4B, and grain filling duration on 2B. The higher number of QTLs identified in BIL2 for heat stress tolerance suggests the importance of assessing the effects of intraspecific variation of *Ae. tauschii* in wheat breeding as it could modulate the heat stress responses/adaptation. Our study provides useful genetic resources for uncovering heat-tolerant QTLs for wheat improvement for heat stress environments.

## 1. Introduction

With the rising global demand for food and frequent occurrences of abiotic stresses, such as heat stress, it is crucial to develop wheat varieties with high yield potential and heat tolerance [1,2]. Therefore, breeding for heat stress tolerance is essential to enhance wheat grain yield and adaptation. Typically, crop breeders utilize genetic variation to improve crops against environmental stress. However, due to intensive selection for grain yield and other desirable traits, most wheat varieties exhibit narrow genetic diversity, which limits their potential for further improvement of heat stress tolerance [3]. Exploring and utilizing the adaptive potential of wild progenitors in a systematic approach is a powerful strategy and a practical solution to identify promising material and broaden the genetic base of wheat [4].

Wild relatives of wheat, including the Aegilops species, are valuable resources for developing new genetic materials. Numerous studies have reported the tolerance of the species to abiotic stresses [5,6,7]. Among the Aegilops species, *Aegilops tauschii* is the most promising due to the similarity of its D genome to that of bread wheat. No special cytological technique is required to induce meiotic recombination [6,8,9].

Globally, attempts have been made to broaden the genetic diversity of wheat utilizing *Ae. tauschii.* Several QTLs and marker–trait associations (MTAs) have been identified for abiotic stress-adaptive traits [10,11,12,13]. However, a limited number and diversity of *Ae. tauschii* has been explored and utilized in the development of synthetic wheat. To ensure systematic exploration and extensive utilization of the tremendous diversity of *Ae. tauschii* in the breeding programs, multiple synthetic derivatives (MSDs) have been developed, utilizing 43 *Ae. tauschii* accessions [8,14]. The MSD population was systematically evaluated to identify useful stress-adaptive traits, MTAs, and QTLs. The MSD population displayed significant genetic diversity when exposed to heat, drought, and heat–drought combinations under field conditions in Sudan. Several MTAs were identified to be associated with heat, drought, heat–drought, and grain characteristics and quality [15,16,17,18,19,20].

The MSDs, as a mixture population of several BC_1_F_7_ lines derived from crosses of 43 *Ae. tauschii* accessions, are expected to have high linkage disequilibrium; therefore, the previously identified MTAs would not be useful for direct use in breeding unless validated. A better approach for efficient and precise mapping and validation of the QTLs is to reduce the genomic contribution of *Ae. tauschii* in the progeny by utilizing one or more backcrosses and developing backcross-derived inbred lines (BILs). Therefore, we developed BILs targeting traits associated with heat stress tolerance to be evaluated in field experiments under heat and normal conditions and select lines fixed for desired donor alleles of QTLs.

We selected two MSD lines previously identified by Elbashir et al. [21] based on their high heat tolerance efficiency. Two backcross recombinant inbred lines (BILs) were developed utilizing the primary synthetic derivatives of the two MSD lines crossed and backcrossed to ‘Norin 61′ (N61). Then, crosses were advanced to BC_1_F_5_ using the single-seed descent method. 

The two BILs were used in this study to identify QTLs associated with heat stress tolerance under field conditions. We evaluated the heat stress response of the two BIL populations (107 lines from BIL1 and 164 lines from BIL2) across four environments in Sudan with temperature gradients ranging from relatively cool temperature in the north (Dongola), to a continuous heat stress condition in the central region (Wad Medani). Utilizing genotyping by random amplicon sequencing direct (GRAS-Di) markers, we identified novel QTLs for heat stress tolerance and verified the MTAs previously reported using the MSD population. Our study offers novel genetic resources and QTLs for breeding wheat with enhanced tolerance to heat stress conditions.

## 2. Results

### 2.1. Climate Conditions during the Growing Seasons

Temperature data were recorded in season 2020/2021 at Wad Medani (WM1) and in season 2021/2022 at Dongola (DN), Waha (WA), and Wad Medani (WM2). The mean maximum and minimum temperatures for each environment during the growing seasons were 30.0 and 11.6 °C at DN, 32.9 and 16.4 °C at WA, 36.3 and 17.9 °C at WM1, and 35.7 and 15.9 °C at WM2, respectively (Figure 1a–d). In general, DN was the coolest among the four environments during the growing season (Figure 1a), whereas WM1 was the hottest. During the grain-filling period, the maximum temperature ranged from 21.5 to 36.8 °C at DN, 25.0 to 43.0 °C at WA, 27.0 to 44.0 °C at WM1, and 27.0 to 42.0 °C at WM2. At DN, maximum temperatures ≥40.0 °C during the grain-filling period were not reported; however, at WA, WM1, and WM2, 7, 19, and 11 days were reported with maximum temperatures ≥40.0 °C, respectively. Likewise, the number of days with mean temperatures ≥20.0 °C during the grain filling period was 33, 57, 75, and 67 days at DN, WA, WM1, and WM2, respectively.

### 2.2. Impact of Heat Stress on the BIL Populations 

The analysis of variance (ANOVA) revealed significant effects of genotype, environment, and their interaction on most traits in both BILs, with the exception of genotypic effect on plant height (PH) and biomass (BIO) in BIL1, environmental effect on thousand kernel weight (TKW) in BIL1, and the genotype and environment (G × E) interaction on PH in BIL2 (Appendix A).

In BIL1, the mean grain yield (GY) was 5553 kg ha^−1^ in Dongola (DN), 2340 kg ha^−1^ in Waha (WA), 2426 kg ha^−1^ in Wad Medani first season (WM1), and 5211 kg ha^−1^ in Wad Medani second season (WM2). The mean GY in BIL2 was 5408 kg ha^−1^ in DN, 2531 kg ha^−1^ in WA, 2451 kg ha^−1^ in WM1, and 6136 kg ha^−1^ in WM2. The GY, BIO, days to heading (DH), days to maturity (DM), and PH were significantly reduced at WM1, and WA compared to DN and WM2 in both BILs (Appendix A). 

High-to-moderate broad-sense heritability (h^2^) estimates were found for GY, DH, and DM in both BILs (Appendix A). Moderate-to-low h^2^ estimates were found for GFD, BIO, and HI in both BILs. The h^2^ estimated for PH and TKW varied between the two BILs (Appendix A).

We calculated the two stress tolerance indices (STIs) for GY. The STI1-GY was calculated taking DN as the non-stress environment, whereas the STI2-GY was calculated considering WM2 as a non-stress environment. Then, we performed regression analysis between the GY at DN and the STI1-GY at WA, and the STI1-GY at WM1 (Figure 2a,b). We also performed a regression analysis between the GY at WM2 and the STI2-GY at WM1 (Figure 2c). In BIL1, six genotypes exhibited higher STI values than the recurrent parent, N61, in both WA and WM1 (Figure 2a,b). In BIL2, five genotypes exhibited higher STI and GY values than N61 at WA. At WM1, three genotypes exhibited higher STI and GY values than N61 (Figure 2a,b). For STI2, sixteen and seven genotypes in BIL1 and BIL2, respectively, exhibited higher STI2 values relative to N61 (Figure 2c). Three genotypes in both BILs showed higher GY and STI2 values than N61 (Figure 2c). 

### 2.3. Relationship among Traits

In both BILs, GY significantly and consistently correlated (*p* < 0.05) with BIO and HI in all environments (Appendix A). However, at DN, GY significantly correlated with seed number per spike (SN) and TKW in BIL1 and BIL2, respectively. At WM1, GY significantly correlated with SN in BIL1 and BIL2, whereas at WM2, GY correlated only with TKW in BIL2. The STI1-GY at WA and WM1 significantly correlated with BIO and HI in both BILs. Likewise, the STI2-GY at WM1 significantly correlated with SN, BIO, and HI in both BILs (Appendix A). 

### 2.4. Linkage Maps for the BILs

Both BIL1 and BIL2 were genotyped using genotyping by random amplicon sequencing direct (GRAS-Di). The details of the linkage map construction of BIL1 were mentioned in Ahmed et al. [22]. Briefly, the high-density linkage map was developed using 2882 markers. The markers were distributed unevenly across multiple chromosomes and subgenomes. The D-subgenome recorded the highest marker density followed by B- and A-subgenomes. Chromosome 3D displayed the highest number of markers, whereas chromosome 6B showed the lowest number of markers. 

In BIL2, 19,765 markers were used for GRAS-Di genotyping, with 6504 exhibiting polymorphisms between the synthetic wheat donor parent, Syn44, and the recurrent parent, N61. Of these polymorphic markers, 3404 (52.3%) were of high quality, with an average of 162 markers per chromosome. A high-density linkage map was constructed utilizing the 3404 markers spread across 21 linkage groups, covering a total genetic distance of 5673.33 cM (Figure 3 and Appendix A). The average distance per chromosome was 270.16 cM. The markers were not uniformly distributed among the chromosomes and subgenomes. Most markers were mapped to the B- (1207, 35.46%) and D- (1201, 35.28%) subgenomes, which had total genetic lengths of 1978.99 and 1943.64 cM, respectively. A total of 996 (29.26%) markers were mapped to the A-subgenome with a total genetic length of 1750.7 cM (Appendix A). The B- and D-subgenomes had the highest marker density, with one marker per 1.7 cM, whereas the A-subgenome had one marker per 2.0 cM (Appendix A). Chromosome 7D had the highest number of markers (264) with a genetic distance of 435.36 cM, whereas chromosome 5A had the lowest number of markers (69) with a genetic distance of 207.37 cM. Chromosomes 2A, 1B, and 2B had marker gaps greater than 30 cM (Appendix A).

### 2.5. Identified QTLs in All Environments

QTL analysis was performed in both BILs using DH, DM, GFD, PH, GY, BIO, TKW, HI, and SN. In addition, stress tolerance indices for GY, BIO, and TKW and their Best Linear Unbiased Predictor (BLUP) were used. Using inclusive composite interval mapping of QTLs with additive and dominance effect analysis (ICIM-ADD), we identified 40 QTLs in BIL1 for the studied traits in the four environments. The identified QTLs were mapped on all chromosomes except 3A, 4A, 5A, 3B, 4B, and 6B. The LOD scores of the identified QTLs ranged from 2.50 to 5.22, and the phenotypic variation explained ranged from 5.14 to 15.43%. The numbers of QTLs detected at WM1, DN, WA, and WM2 were seventeen, ten, nine, and four, respectively. The highest number of QTLs was identified in the D-subgenome (17, 42.5%) for DH, GFD, GY, HI, PH, STI1-GY, STI1-TKW, STI2-TKW, and TKW. A total of 13 QTLs (32.5%) were identified in the A-subgenome for BIO, DH, GY, PH, STI1-BIO, STI1-TKW, STI2-TKW, and TKW, whereas 10 QTLs (25%) were identified in the B-subgenome for DM, GFD, HI, PH, SN, STI1-TKW, and TKW. The QTLs associated with GY were detected on chromosomes 6A, 1D, 3D, and 5D, whereas QTLs of both STI-GYs were detected on chromosomes 1B, 1D, and 6D (Appendix A).

In BIL2, 153 QTLs were identified for the studied traits in the four environments. The identified QTLs were mapped on all chromosomes except 1B and 1D. The LOD score of the identified QTLs ranged from 2.50 to 18.63, and the phenotypic variation explained ranged from 0.32 to 18.91%. The highest number of QTLs was identified at WM1 (72), whereas 28, 27, and 26 QTLs were detected at WA, DN, and WM2, respectively. Fifty-six (36.60%) QTLs were identified in the A-subgenome associated with all studied traits, while fifty-four (35.29%) QTLs were identified in the D-subgenome for BIO, DH, DM, GFD, GY, HI, PH, STI2-BIO, STI2-GY, STI1-TKW, STI2-TKW, and TKW. The lowest number of QTLs (43, 28%) was identified in the B-subgenome associated with BIO, DH, DM, GFD, HI, PH, STI1-BIO, STI2-BIO, STI2-GY, and TKW. The QTLs of GY were detected on chromosomes 5A, 7A, and 5D, while those associated with both STI-GYs were detected on chromosomes 3A, 5A, 4B, and 6D (Appendix A).

### 2.6. QTLs Associated with Heat Stress Response in Both BILs

In BIL1, 14 QTLs associated with stress tolerance indices for GY, BIO, and TKW were detected on chromosomes 1A, 2B, 1D, 5D, and 7D (Table 1, Figure 4). Three QTLs for STI1-GY were only detected in WA on chromosomes 1B, 1D, and 6D, explaining a phenotypic variation of 7.50, 15.43, and 8.67, respectively. A single QTL associated with STI1-BIO was detected in WM1 on chromosome 1A, explaining 5.14% of the phenotypic variation. For STI1-TKW and STI2-TKW, 10 QTLs were detected in both heat stress environments (WA and WM1) on chromosomes 1A, 2B, 1D, 5D, and 7D, with phenotypic variation ranging from 8.29 to 14.54%. Of these ten QTLs, two on chromosome 1A and one on chromosome 2B were stable and overlapped across the two environments. One of the two stable QTLs of STI1-TKW on chromosome 1A was detected at 173 cM between the flanking markers AMP0035547 and AMP0004300, and the phenotypic variations explained were 9.89 and 14.54% at WA and WM1, respectively. The other QTL was detected at 118–119 cM between the flanking markers AMP0036610 and AMP0034796 and the phenotypic variations explained were 11.37 and 14.37% at WA and WM1, respectively. Meanwhile, the QTL of STI1-TKW on chromosome 2B was detected at 62 and 63 cM between the flanking markers AMP0009891 and AMP0006464 and explained 8.66 and 10.41% of the phenotypic variation at WA and WM1, respectively (Table 1, Figure 4).

In BIL2, 25 QTLs associated with stress tolerance indices for GY, BIO, and TKW were identified on chromosomes 3A, 5A, 2B, 4B, and all D-subgenome chromosomes, except for chromosome 1D. Six QTLs associated with STI1 or STI2 for GY were found on chromosomes 3A, 5A, 4B, and 6D. Among the six QTLs, a QTL was stable and consistently identified at WA and WM1 at 88 cM on chromosome 5A between the flanking markers AMP0011577 and AMP0030240, explaining 10.49 and 13.70% of the phenotypic variation, respectively (Table 1 and Table 2). For STI-BIO and STI2-BIO, 12 QTLs were detected on chromosomes 3A, 5A, 2B, 3D, 4D, 5D, and 7D. Among them, one QTL on chromosome 5A was stable. The stable QTL was detected at 3 cM between flanking markers AMP0003832 and AMP0029058 and explained 3.10 and 6.20% of phenotypic variations. Seven QTLs were detected at WM1 associated with STI1 and ST2 for TKW on chromosomes 3A, 2D, 4D, and 6D (Table 1).

Chromosomes 1A, 2A, 7A, 2B, 5B, and 7B, as well as all D-subgenome chromosomes except 7D, were common regions that harbored different QTLs in both BILs. BIL1 exhibited a higher number of QTLs on chromosomes 1A, 2B, and 1D, whereas BIL2 had a greater number of QTLs on all D-subgenome chromosomes, except 1D and 7D, as well as 7A and 7B (Figure 4). 

Interestingly, some of the identified QTLs in the two BILs co-located in the same region of the chromosome with some MTAs identified in previous studies conducted using the related population of the MSDs (Figure 4). In addition, most of the co-located QTLs in both BILs were identified in the D-subgenome with the chromosome 5D contributing the most (Figure 4).

## 3. Discussion

In the face of climate change, evaluating wheat lines with limited genetic diversity or with similar genetic backgrounds to the elite breeding lines would not be useful for delivering superior climate-smart varieties. Broadening genetic diversity and bringing more useful alleles and genes from wild relatives of wheat has proven to be successful [6,7,10,27,28]. Therefore, a population with a wide diverse genetic background has been developed utilizing 43 *Ae. tauschii* accessions and named multiple synthetic derivatives (MSDs) [8,14]. The MSD population was evaluated over several seasons in multiple environments ranging from normal to continuous heat stress conditions. The results showed that a number of superior lines could be identified under various conditions and for different climate-resilient traits [15,16,17,18,21]. Moreover, GWAS analysis identified several MTAs under heat, drought, and heat–drought conditions [15,16,17,19,20]. Following the identification of several heat-tolerant lines from the MSD population, backcross inbred lines (BILs) were developed in the background of those heat-tolerant lines with exotic alleles from two *Ae. tauschii* accessions collected from two different locations. 

In this study, we evaluated two BILs comprising 271 lines in four environments ranging from relatively cooler environments at Dongola in northern Sudan to continuous heat stress environments at Wad Medani in central Sudan. The analysis of variance showed that the lines of the two BILs exhibited significant variation in grain yield and some agronomic traits under the relatively cool and continuous heat stress conditions, confirming those previously reported for the MSD population and other bread wheat genotypes carrying exotic alleles [10,15,21]. The great variation in key traits such as grain yield coupled with a high broad-sense heritability demonstrated the possibility of exploring them in breeding for heat stress tolerance.

In this study, heat stress reduced grain yield compared to the relatively cooler environment at DN, especially at WA and WM1. Compared to DN, the mean grain yield in BIL1 reduced by about 58% at both WA and WM1. Similarly, in BIL2, reductions in GY of 52 and 56% were found at WA and WM1, respectively, compared to that at DN. However, the mean grain yields of the two BILs at WM2 were always comparable with those at DN despite the fact that significant reductions in the grain yield of some individual lines were observed. Figure 1 shows that the two BILs had longer grain-filling periods at the two WM environments than DN, although the heading time of the two BILs was earlier at the WM environments. This might be due to the significant G × E interactions observed for different traits, especially DH, DM, GFD, GY, and BIO (Appendix A). For instance, about 72% of the BIL1 lines that showed a reduction in grain yield at WM2 compared to DN were either early (with DH less than 60 days) or late (DH more than 70 days). This might be related to the vernalization gene doses in these populations, which deserve further investigation.

Compared to the recurrent parent, N61, several lines in both BILs showed better performance under both stressed and non-stressed environments. In both BILs at the heat stress environment of WM1, twelve lines exhibited higher STI-GY, out of which seven lines had higher grain yield than N61. Similarly, at WA, sixteen lines showed higher STI-GY than N61, of which eight lines also had higher grain yield. The high grain yield and stress tolerance index were closely associated with the accumulation of high biomass as indicated by the strong correlation of grain yield and STI-GY with the biomass at both heat-stressed environments. High biomass accumulation was found to be important under the same heat stress environments [29], as well as when exotic-derived lines from *Ae. tauschii* were compared under heat stress with elite wheat lines [10,30]. The high-yielding heat-tolerant lines identified in both BILs are promising and could be integrated into wheat heat stress tolerance breeding programs.

### 3.1. The High-Resolution Linkage Maps 

The GRAS-Di platform generates many chromosome-spanning genetic markers, allowing the construction of high-resolution linkage maps [31]. We constructed two genetic maps using 2882 and 3404 GRAS-Di markers in BIL1 and BIL2, respectively, that were polymorphic between the synthetic parents and N61. The map lengths in both BILs were relatively long, possibly because of the high density of GRAS-Di markers, which generally increases the total length of the linkage map [31]. Nevertheless, the length of our maps was comparable to those reported in wheat [32,33]. The D-subgenome was longer in BIL1 than that of the A- and B-subgenomes [22]. However, in BIL2, the D-subgenome map was similar in length to that of the B-subgenome. The similar map length of the B- and D-subgenomes in BIL2 might indicate higher polymorphism between N61 and the synthetic donor of BIL2 (Syn44) than that of BIL1 (Syn32). Additionally, the higher number of high-quality markers used for map construction in BIL2 than in BIL1 suggests that the D-subgenome of Syn44 exhibited higher interaction with the A- and B-subgenomes of N61. The D-subgenome has been reported previously as the shortest with the lowest number of markers [34,35] due to its low level of polymorphism [36]. Unlike these reports, in this study, a higher polymorphism was recorded in the D-subgenome, which might be attributed to its origin from the wild D-genome introduced from *Ae. tauschii*. Therefore, the two BILs used here could provide a unique opportunity to study the D-subgenome and could be a valuable breeding material for climate change-resilient wheat improvement.

### 3.2. QTLs Identified in All Environments

QTL analysis performed using the two BILs identified 193 QTLs on all chromosomes except 3A, 4A, 5A, 3B, 4B, and 6B under stress and non-stress environments. Although QTLs were detected for all traits studied, the highest number of QTLs was associated with TKW and PH (26 QTLs each), followed by the phenological traits. A total of nine QTLs for grain yield were identified in both BILs. A total of 156 QTLs (80.8%) were identified under the heat stress environments of WA, WM1, and WM2. The high number of QTLs identified under heat stress could be a good indicator of the usefulness of such populations in mining QTLs associated with stress-adaptive traits, as has been previously reported [15,37,38]. The D-subgenome contributed 36.9% of the total QTLs identified, including five out of nine GY QTLs and thirteen out of forty QTLs associated with the stress tolerance index. Our results confirmed the importance of the D-subgenome derived from *Ae. tauschii* for mining stress-adaptive traits [18].

As shown in Figure 4, a number of the QTLs identified in this study were co-located with some of the previously reported QTLs under heat stress conditions on chromosomes 1A, 2A, 2B, 5B, 7B, 1D, and 2D [23,24,37,39,40,41,42,43]. For instance, a number of STI-TKW QTLs were co-localized with previously identified QTLs for HSI-TGW (heat susceptibility index of thousand-grain weight) and TKW under heat and combined heat–drought on chromosomes 1A, 2B, and 2D.

#### 3.2.1. Identification of Stable Major QTLs for Yield- and Heat Stress Tolerance-Related Traits

The present study identified a total of 39 QTLs related to heat stress tolerance in both BILs, of which 25 were from BIL2. The higher number of QTLs identified in BIL2 for heat stress tolerance suggests that BIL2 has greater heat adaptability than BIL1. When the results of this study are linked with that reported by Mahjoob et al. [44] on leaf hair density in two intra-specifically diverse *Ae. tauschii* accessions, it is clear that different traits have evolved independently in different lineages of the species. Apparently, KU-2124, collected from TauL2 in Iran, contributed more heat stress adaptation traits than KU-2039, collected from TauL1 in Afghanistan. Thus, to further improve the resilience of wheat to climate change, the intraspecific diversity and lineage differences of *Ae. tauschii* should be explored and exploited. 

We used STIs for GY, BIO, and TKW and identified new QTLs for heat stress tolerance, including STI-GY on chromosomes 1B, 4B, 1D, and 6D; STI-TKW on chromosomes 3A, 2B, and 4D; and STI-BIO on chromosomes 1A, 3A, 3D, 4B, 4D, 5A, and 7D in both BILs. Recently, Farhad et al. [45] reviewed QTLs identified under heat stress in wheat and reported several QTLs related to heat stress tolerance in wheat identified so far. Several QTLs for heat and stress susceptibility indices (HSI, SSI) for kernel weight, grain-filling duration, and grain yield have been identified. For instance, QTLs for SSI-GY on chromosomes 2A, 5A, and 5B [25,26] and HSI-TKW on chromosomes 1A, 2B, 2D, 5B, 5D, 6A, 6B, 6D, and 7D [23,24,26] have been reported. These newly identified heat stress QTLs have the potential to improve GY under heat stress after further validation.

Out of the 39 heat stress-related QTLs, 12 were found to be major QTLs due to their high contribution to phenotypic variation (10.1–15.5%). The 12 major QTLs were detected on chromosomes 1A, 3A, 5A, 2B, 1D and 5D. Out of the twelve QTLs, eight were for STI-TKW on chromosomes 1A, 3A, 2B, 1D, and 5D. Three of these QTLs were stable across two to three environments and in the BLUP. Additionally, four QTLs were found for STI-GY on chromosomes 3A, 5A, 1D, and 6D with one stable across two environments (Table 2 and Appendix A). Following the validation of these major stable QTLs, they could be employed to bolster the heat tolerance of wheat with the aid of marker assistance selection. 

#### 3.2.2. Common and Specific Regions of Detected QTLs in BIL1 and BIL2

The QTLs identified under heat stress conditions on chromosomes 7A and 7B for DH, DM, GFD, GY, and TKW were specific to BIL2. We found a polytropic effect for DM, DH, and PH under heat stress on chromosome 7A, and for DM and TKW on 7B. A QTL for SSI-KW was co-localized with QTLs of TKW on chromosome 7B [40]; however, a QTL associated with heat stress was not previously reported on chromosome 7A. Thus, these regions in 7A and 7B may be necessary for improving DH, DM, PH, and TKW for heat stress adaption. 

The uniqueness of BIL populations used in this study was demonstrated by QTLs identified on chromosomes 3D, 4D, 5D, and 6D. We also found a pleiotropic effect between several important traits in the four D-chromosomes. To our knowledge, no heat stress-related QTLs have been reported in these regions of the chromosomes. Previously, by utilizing the MSD population, several MTAs have been identified in these regions [15,16,17], which were colocalized with the QTLs identified in this study. This implies that QTLs in these regions are novel and could be specific to the exotic alleles introduced from *Ae. tauschii* in the MSD and BIL populations. 

On chromosome 6D, we identified two QTLs for STI-GY in BIL1 and BIL2 at 53 and 92 cM, respectively. The close position of the two QTLs suggests that they might be the same. No QTL associated with STI has been reported in this region; thus, it is mostly a novel QTL. Following validation of this QTL within the identified hot-spot regions, it could be utilized to enhance the heat tolerance of wheat via marker-assisted selection (MAS).

The pleiotropic and new QTLs identified on chromosomes 7A, 7B, 3D, 4D, 5D, and 6D suggest that these regions are valuable for heat stress tolerance-related traits. Therefore, they could be targeted and prioritized for mining heat stress tolerance alleles and genes and marker-assisted selection. Utilizing the MSD population, we demonstrate the efficacy of the systematic approach we employed in establishing the BIL population.

In this study, we conducted QTL analysis for agronomic traits under heat stress and non-heat stress conditions using two BIL populations. We identified 39 novel QTLs associated with GY, BIO, and TKW stress tolerance indices. More importantly, we identified nine novel STI-GY QTLs on chromosomes 3A, 5A,1B, 4B, 1D, and 6D. Notably, we found a critical region in chromosome 6D where we identified an STI-GY QTL in both BILs, located only 39 cM apart. This region might be of high significance and deserves attention for potential targeted marker development. We also discovered new QTLs associated with important agronomic traits, including grain yield and thousand-kernel weight under heat stress conditions. These QTLs could be employed for marker-assisted selection and gene discovery after validation and can be given priority in breeding for heat stress tolerance. 

## 4. Materials and Methods

### 4.1. Plant Materials

In this study, we used two populations of backcrossed recombinant inbred lines (BILs, BC_1_F_5_). Through intensive evaluation of multiple synthetic derivatives (MSDs), developed utilizing 43 accessions of *Ae. tauschii* and the durum wheat cultivar ‘Langdon’, then crossed and backcrossed with the hexaploid wheat cultivar N61 [8,14], under field heat stress conditions, two heat-tolerant MSD lines were identified [21]. We utilized the two *Ae. tauschii* parents (KU-2039 and KU-2124) of the two synthetic derivatives (Syn32 and Syn44) for the development of the two BILs. KU-2039 was collected from Afghanistan, whereas KU-2124 was collected from Iran. The two accessions were classified into two distantly related lineages, TauL1 for KU-2039 and Taul2 for KU-2124. 

First, the two *Ae. tauschii* accessions (DD genome) were crossed with the durum wheat cultivar ‘Langdon (AABB genome)’ to develop two primary synthetic hexaploid wheats (SHWs). Then, the two SHWs were crossed and backcrossed with N61 to generate BC_1_F_1_. The single-seed descent (SSD) method was used to develop the two BILs (BC_1_F_5_). Initially, the first BIL, generated from KU-2039 (BIL1) consisted of 166 lines, whereas the second BIL, generated from KU-2124 (BIL2), consisted of 236 lines. All lines with very late headings or those that failed to flower due to the vernalization requirement were excluded after evaluation under the heat stress conditions at Wad Medani, Sudan. The final numbers used for the multi-environment evaluation in BIL1 and BIL2 were 107 and 164 lines, respectively.

### 4.2. Experimental Sites and Design

We conducted field experiments in two seasons (2020/21 and 2021/22) in Sudan. In the first season, the experiment was conducted at the Gezira Research Station Farm (GRSF), Agricultural Research Corporation (ARC), Wad Medani (14°24′ N, 29°33′ E, 407 m a.s.l. abbreviated as WM1). In the second season, in addition to GRSF (abbreviated as WM2), the experiment was conducted at Dongola Research Station Farm, ARC, Dongola, Northern Sudan (19°08′ N, 30°27′ E, 239 m a.s.l.; abbreviated as DN) and Waha Farm, south of Khartoum (32°33′35.6328″ E, 380 m a.s.l.; abbreviated as WA). For more details on the characteristics of the experimental sites at DN and GRSF, refer to Elbashir et al. [23] and Mohamed et al. [19]. The Waha site is characterized as a semi-arid area with mostly haplusterts soil type, clay loam in texture, with a pH range of 7.5–8.0, low N (0.05–0.20%), and low available P (30–60 kg ha^−1^) (http://susis.sd/south-region-0-30/, accessed on 12 November 2023). All experiments were arranged in an augmented randomized complete block design with five blocks and six checks (five Sudanese cultivars and the recurrent parent, N61. 

At all sites, seeds of each genotype were manually sown in a plot consisting of two rows, 1.0 m long and 0.2 m apart. At WM1 and WM2, sowing was performed during the 4th week of November, whereas at DN and WA, sowing was performed during the 2nd week of December. All crop management including seed treatments, irrigation, fertilizer application, weed control, etc., followed the recommendations of the ARC for wheat production in Sudan as previously described [19,21,46]. During the winter season, there is usually no rainfall at any of the experimental sites. Therefore, wheat is fully irrigated with irrigation intervals of 10–14 days, depending on weather conditions, mainly temperature, to avoid water stress.

### 4.3. Phenotyping of BIL Populations

#### 4.3.1. Trait Evaluation

At all sites, phenological, yield, and some yield component traits were recorded. Days to heading (DH) was recorded as the number of days from first irrigation until 50% of the spikes merged from the leaf sheath. Days to maturity (DM) was the number of days from first irrigation until 90% of the plants had lost the green color of the glumes. Grain-filling duration (GFD) was calculated as the interval in days between DH and DM. Plant height (PH, cm) was taken at maturity by measuring the distance from the ground to the top of the spike, excluding awns.

Grain yield (GY) and related traits, including biomass (BIO), thousand-kernel weight (TKW), harvest index (HI), and seed number/spike (SN) were recorded. GY and BIO were determined as grain weight and above-ground dry weight per plot, respectively, and then converted to kg ha^−1^ for further analysis. TKW (g) and SN were determined from random samples of 10 spikes. HI was measured as the ratio of BIO to GY (GY/BIO × 100).

To identify heat stress-tolerant genotypes, the stress tolerance index (STI) for GY (STI-GY), BIO (STI-BIO), and TKW (STI-TKW) was calculated according to Fernandez [47] as
$$STI-GY=\frac{\left(Y_{N})(Y_{S}\right)}{{\left(Y_{\bar{N}}\right)}^{2}}$$
where $Y_{N}$ is GY under non-stress cconditions $Y_{S}$ is GY under heat stress conditions, and $Y_{\bar{N}}$ is the mean GY under non-stress conditions. For BIO and TKW, GY was replaced with BIO and TKW, respectively. 

We calculated STI-GY twice. First, STI1-GY was calculated regarding the GY values at DN as the non-stress environment and WA or WM1 as the hot environment. Second, STI2-GY was calculated considering the GY of WM2 as the non-stress environment and WM1 as the stress environment.

#### 4.3.2. Statistical Analysis of Phenotypic Data

Analysis of variance (ANOVA) for all traits was performed in Plant Breeding Tools v. 1.4.2 [48]. The genotype (G), environment (E), and (G × E) interaction were considered random effects. For each trait, the Best Linear Unbiased Predictor (BLUP) was computed and subsequently employed for additional analysis. We used Tukey’s honestly significant difference (HSD) test for between-environment comparisons and the *t*-test for between-BIL comparisons. Pearson’s correlation coefficient among traits in each environment was calculated using IBM SPSS Statistics for Windows v. 28.0.1.1(15) [49]. Broad-sense heritability was estimated in Plant Breeding Tools.

### 4.4. Genotyping of the BILs, Map Construction, and QTL Analysis

#### 4.4.1. DNA Extraction

Total genomic DNA was extracted from 2-week-old leaves of BILs using the CTAB method [50]. The DNA samples (20 µL; 50–100 ng µL^−1^) were sent to Eurofins Genomics Company, Tokyo, Japan (https://eurofinsgenomics.jp/jp (accessed on 20 October 2023)), for a whole-genome scan with genotyping by random amplicon sequencing direct (GRAS-Di) markers.

#### 4.4.2. Maps Construction 

The linkage map for both BILs was constructed using GRAS-Di markers. The construction of GRAS-Di libraries followed the protocol outlined in Hosoya et al. [51]. The libraries were sequenced using the Illumina HiSeq series. GRAS-Di software (TOYOTA, Aichi, Japan, https://eurofinsgenomics.jp/jp/service/ngs/gras-di/ (accessed on 21 January 2024)), which is commercially accessible, was used for genotyping. The software assesses the quality of markers by applying empirical criteria for genotyping reproducibility [31]. This assessment is based on the number of reads and the consistency of genotyping results across samples (presence and absence of the reads). To determine reproducibility, the software uses trial data from the GRAS-Di database for crop species (Patent ID P2018-42548A). The markers are then ranked using A, B, C, D, and E labels, with A representing the highest reproducibility (≥99.99%), B reproducibility between 99.98% and 99.99%, C reproducibility between 99.90% and 99.98%, and D reproducibility between 99.80% and 99.90%. The A-, B-, C-, and D-ranked markers are of high quality to be used for genotyping. Because the E-ranked markers contain tested samples with missing values, they are not recommended to be used for genotyping [31].

Previously, we genotyped the BIL1 population using GRAS-Di markers and used this material for QTL analysis of seed dormancy [22]. In this study, we genotyped BIL2 using the same method as BIL1. Therefore, both BILs’ genetic maps were utilized in the current study. The details of BIL1 map construction have been mentioned in Ahmed et al. [22]. In the same way, the 164 lines of BIL2 were genotyped with 19,765 GRAS-Di markers. We removed markers amplified in all samples from all parents, markers of low quality (E), and markers with at least one mismatch. The remaining 6504 markers were used to construct the linkage maps. In the first step, we implemented the BIN tool algorithm in the IciMapping software version 4.2 [51]. The remaining markers were binned according to their segregation patterns. After binning, we grouped the markers using a logarithm of odds (LOD) threshold of 3.0 [52]. Linkage groups were assigned according to the genomic position of the SNP markers determined during SNP calling. Recombination frequencies between markers were converted to centiMorgans (cM) using the Kosambi mapping function [53]. We inspected the initial linkage map for duplicate lines, segregation distortion, switched alleles, and single and double crossovers (genotyping errors) using the R/qtl and R/ASMap packages available in the R Statistical Computing Environment v 2.15.1 [54,55]. Finally, after removing low-quality markers and correcting for genotyping errors, the genotypic data of BIL1 and BIL2 with 2882 and 3404 high-quality markers, respectively, were used to construct the final QTL map in IciMapping software version 4.2.

#### 4.4.3. QTL Analysis

The studied traits for BIL1 and BIL2 were used for QTL mapping in QTL IciMapping. We used the mean of the traits at each environment as well as their BLUPs. An inclusive composite interval mapping of QTL with additive and dominance effect (ICIM-ADD) analysis was performed. The significant LOD threshold (2.5) for declaring a QTL (α = 0.05) was determined from 10,000 permutations. We reported a LOD score of significant QTLs ranging from 2.5 to 5.2 in BIL1 and 2.5 to 18.6 in BIL2.

## 5. Conclusions

The systematic exploration, utilization, and evaluation of synthetic wheat developed using 43 *Ae. tauschii* accessions enabled the identification of several heat-tolerant lines and MTAs associated with important heat adaptation traits. In this study, we utilized *Ae. tauschii* donors of two heat-tolerant MSD lines to develop two BILs. The evaluation of the two BILs under field conditions with continuous heat stress revealed 16 heat-tolerant lines as well as 39 novel QTLs associated with heat stress. Specifically, the use of the two BILs permitted us to pinpoint desirable QTLs absent in elite wheat cultivars. The major and stable QTLs identified in this study, associated with the stress tolerance indices of GY and TKW, hold considerable promise and warrant further investigation to fully elucidate their potential applications. Efforts are underway to develop near-isogenic lines (NILs) using the selected heat-tolerant BIL lines to validate the identified QTLs and develop markers that will assist in wheat genetic improvement under heat stress.

## Figures and Tables

**Figure 1 plants-13-00347-f001:**
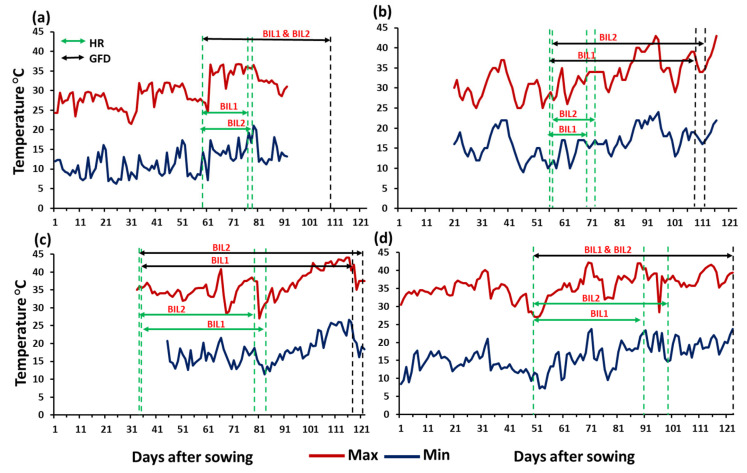
The daily maximum and minimum temperatures recorded during the growing season at (**a**) Dongola (DN), (**b**) Waha (WA), (**c**) Wad Medani first season (WM1), and (**d**) Wad Medani second season (WM2). The heading ranges (HRs, days) and grain filling duration (GFD, days) of both BIL1 and BIL2 are indicated by the dotted green and black vertical lines, respectively.

**Figure 2 plants-13-00347-f002:**
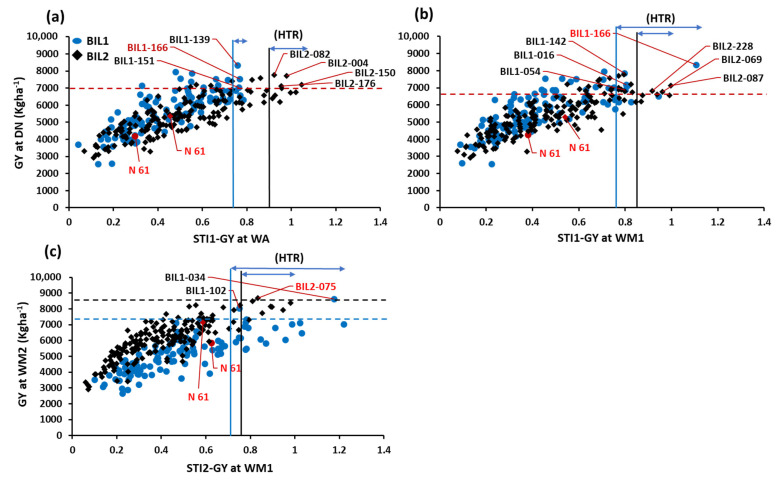
Comparison of grain yield at Dongola (DN) for the two populations (BIL1 and BIL2) using the stress tolerance index (STI). The STI-GY was calculated twice. First, STI1-GY was calculated regarding the GY values at DN as the non-stress environment and WA or WM1 as the hot environment. Second, STI2-GY was calculated considering the GY of WM2 as the non-stress environment and WM1 as the stress environment. The GY at DN was regressed against STI1-GY at (**a**) WA and (**b**) WM1. The GY at WM2 was regressed against STI2-GY at (**c**) WM1. The red circle and diamond represent the recurrent parent N61 in BIL1 and BIL2, respectively. The horizontal dashed red line is plotted based on the GY mean plus the Least Significant Difference (LSD _0_._05_) of both BIL1 and BIL2. Similarly, the horizontal dashed blue line corresponds to the GY mean plus the LSD of BIL1, and the dashed black line reflects the GY mean plus the LSD of BIL2. The vertical blue line is calculated based on the STI mean plus the LSD of BIL1, while the vertical black line is calculated based on the STI mean plus the LSD of BIL2. The horizontal blue arrows represent the heat tolerance ranges (HTR) for both BILs.

**Figure 3 plants-13-00347-f003:**
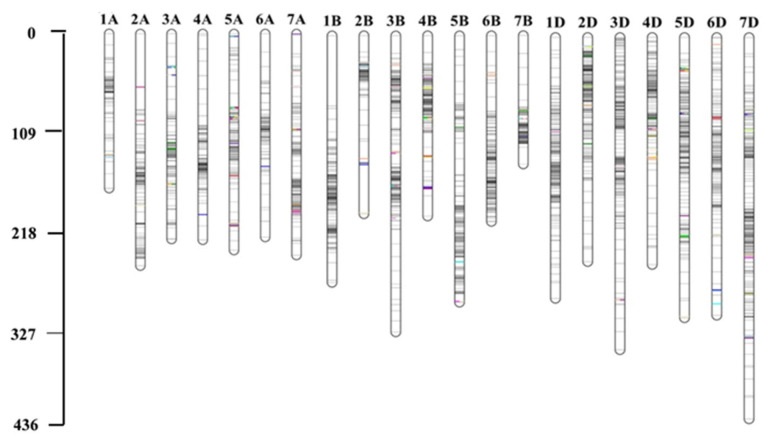
Genetic maps constructed using 3404 GRS-Di markers in BIL2. For BIL1, refer to Ahmed et al. [22]. The positions of the QTLs are colored.

**Figure 4 plants-13-00347-f004:**
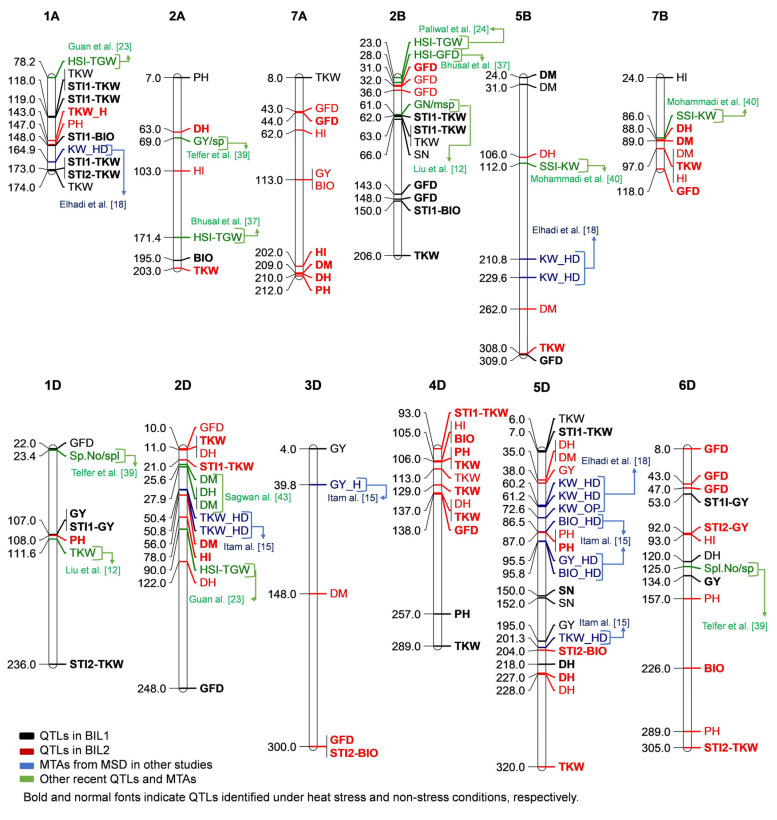
The detected QTLs in common regions in both backcrossed recombinant inbred line populations, BIL1 and BIL2, with co-localized previously reported QTLs. The QTLs detected in BIL1 are in black while those in BIL2 are in red. Normal fonts represent the optimum conditions, while bold fonts indicate the heat condition for both BIL1 and BIL2. The blue color represents the QTLs previously reported in our studies using the multiple synthetic derivative (MSD) line population. The green color represents QTLs reported by other researchers.

**Table 1 plants-13-00347-t001:** The QTLs associated with stress tolerance indices (STIs) of grain yield (GY), biomass (BIO), and thousand-kernel weight (TKW) in two BILs grown in four environments.

Chr ^1^	Trait	Pop ^2^	Pos ^3^ (cM)	Left Marker	Right Marker	LOD ^4^	PVE (%) ^5^	Add ^6^	Co-Localized with
1A	STI1-GY	BIL2(BLUP)	144	AMP0028912	AMP0002760	3.35	8.98	−0.29	
1A	STI1-TKW	BIL1	119	AMP0036610	AMP0034796	4.37	14.37	0.12	Guan et al. [23]
1A	STI1-TKW	BIL1	173	AMP0035547	AMP0004300	2.98	9.89	−0.10	Guan et al. [23]
1A	STI1-TKW	BIL1	118	AMP0036610	AMP0034796	3.74	11.37	0.11	Guan et al. [23]
1A	STI1-TKW	BIL1	173	AMP0035547	AMP0004300	4.27	14.02	−0.11	Guan et al. [23]
1A	STI2-TKW	BIL1	173	AMP0035547	AMP0004300	4.82	14.54	−0.10	Guan et al. [23]
1A	STI1-BIO	BIL1	148	AMP0034796	AMP0020845	2.64	5.14	−0.12	
1B	STI1-GY	BIL1	144	AMP0017578	AMP0023142	2.60	7.49	0.07	
1D	STI1-GY	BIL1	117	AMP0027815	AMP0029085	5.23	15.44	−0.10	
1D	STI2-TKW	BIL1	236	AMP0005955	AMP0027742	3.52	10.14	−0.09	
2B	STI1-TKW	BIL1	63	AMP0009891	AMP0006464	3.21	10.41	0.10	
2B	STI1-TKW	BIL1	62	AMP0009891	AMP0006464	2.95	8.66	0.08	Paliwal et al. [24]
2B	STI1-BIO	BIL2	150	AMP0012513	AMP0026808	2.80	3.46	−0.17	
2D	STI1-TKW	BIL2	21	AMP0020907	AMP0024533	4.11	3.15	0.19	Guan et al. [23]
2D	STI1-GY	BIL2 (BLUP)	152	AMP0017649	AMP0022052	2.51	6.72	−0.07	
3A	STI1-GY	BIL2	49	AMP0010424	AMP0003988	5.33	14.20	0.10	
3A	STI1-TKW	BIL2	129	AMP0014988	AMP0016989	18.63	15.53	−0.19	
3A	STI1-TKW	BIL2	137	AMP0029972	AMP0030211	10.85	8.05	0.14	
3A	STI1-TKW	BIL2	178	AMP0007900	AMP0004728	4.15	2.82	−0.07	
3A	STI2-TKW	BIL2	39	AMP0030786	AMP0010424	2.83	6.29	0.33	
3A	STI1-BIO	BIL2	49	AMP0010424	AMP0003988	3.03	0.57	0.07	
3D	STI2-BIO	BIL2	300	AMP0001446	AMP0012860	4.58	2.97	−0.31	
4B	STI2-GY	BIL2	140	AMP0018665	AMP0020290	3.27	8.54	0.07	
4B	STI1-BIO	BIL2	176	AMP0025189	AMP0026555	3.92	1.12	−0.28	
4B	STI2-BIO	BIL2	178	AMP0026555	AMP0003848	5.10	2.89	−0.36	
4D	STI1-TKW	BIL2	93	AMP0031292	AMP0028457	2.56	1.70	−0.08	
4D	STI2-BIO	BIL2	105	AMP0009857	AMP0007548	4.21	2.01	−0.41	
5A	STI1-GY	BIL2	88	AMP0011577	AMP0030240	3.81	10.50	0.08	Hassouni et al. [25]
5A	STI1-BIO	BIL2	3	AMP0003832	AMP0029058	2.75	6.20	0.20	
5A	STI2-GY	BIL2	88	AMP0011577	AMP0030240	7.84	13.70	0.09	Hassouni et al. [25]
5A	STI2-GY	BIL2	168	AMP0008559	AMP0030185	3.26	5.65	0.06	Hassouni et al. [25]
5A	STI1-BIO	BIL2	3	AMP0003832	AMP0029058	2.94	3.09	0.19	
5A	STI2-BIO	BIL2	87	AMP0025208	AMP0011577	4.14	1.12	0.08	
5A	STI2-BIO	BIL2	227	AMP0001406	AMP0015434	5.74	2.43	−0.39	
5D	STI1-TKW	BIL1	7	AMP0022256	AMP0000398	4.39	14.08	−0.12	Wang et al. [26]
5D	STI2-BIO	BIL2	204	AMP0010296	AMP0028613	2.95	3.10	−0.29	
6D	STI1-GY	BIL1	53	AMP0036794	AMP0032738	3.03	8.67	0.07	
6D	STI2-GY	BIL2	92	AMP0016445	AMP0014713	3.90	6.84	0.07	
6D	STI2-TKW	BIL2	305	AMP0003394	AMP0027092	2.76	5.28	−0.10	Guan et al. [23]
7D	STI2-TKW	BIL1	89	AMP0019618	AMP0017004	2.96	8.30	−0.08	Paliwal et al. [24]
7D	STI2-BIO	BIL2	343	AMP0018976	AMP0002072	2.50	2.56	−0.36	
7D	STI1-GY	BIL2 (BLUP)	100	AMP0021405	AMP0025764	2.51	5.85	0.05	

^1^ Chromosome, ^2^ Populations, ^3^ Position, ^4^ Logarithm of odds, ^5^ Phenotypic variance explained, ^6^ Additive effect.

**Table 2 plants-13-00347-t002:** Key stable/major QTLs overlapped at least in two heat stress environments or the two BILs.

					Environment				
Population	QTL	Chr ^1^	PVE% ^2^	Pos ^3^	DN ^4^	WA ^5^	WM1 ^6^	WM2 ^7^	BIL1/BIL2
BIL1	STI-TKW	1A	11.4–14.4	173		√	√		
	STI-TKW	2B	5.0–10.4	62–63		√	√		
BIL2	TKW	3A	5.9–11.1	40–41	√	√	√		
	STI-GY	5A	10.4–17.2	87–88		√	√		
Both BILs	STI-GY	6D	6.8–8.7	53–92					√

^1^ Chromosome, ^2^ Phenotypic variance explained, ^3^ Position, ^4^ Dongola, ^5^ Waha, ^6^ Wad Madani first season, ^7^ Wad Medani second season.

## Data Availability

Data is contained within the article and Appendix A.

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
