# Peer review of "Heat Stress-Tolerant Quantitative Trait Loci Identified Using Backcrossed Recombinant Inbred Lines Derived from Intra-Specifically Diverse Aegilops tauschii Accessions"

_plants, 2024, doi:10.3390/plants13030347_

Round 1

Reviewer 1 Report

Comments and Suggestions for Authors

Please check the attached file for the comments and suggestions.

Comments on the Quality of English Language

Not applicable

Reviewer 2 Report

Comments and Suggestions for Authors

In line 504, please explain why G, E and GxE are considered as random effects.

The results of anova showed that the GxE is significant. Do you think the QTL mapping should take it into consideration? Or add it in the model?

In relationship among traits, do you think if it is necessary to further dissect the genetic basis behind it?

Comments on the Quality of English Language

 Minor editing of English language required.

Round 2

Reviewer 1 Report

Comments and Suggestions for Authors

attached

Round 3

Reviewer 1 Report

Comments and Suggestions for Authors

The genotypic data were generated from both BIL1 and BIL2 populations. This allowed us to construct the linkage maps for both BIL1 and BIL2. We have published the linkage map of BIL1 and conducted QTL analysis for dormancy (Ahmed et al. 2023). Here in this paper, we genotyped BIL2 and made linkage map for this population. Using these two genotyped populations (BIL1 and BIL2), we conducted phenotypic evaluation for heat-stress tolerance.

For more clarification, we revised Lines 546 – 551.

We hope this explanation clarifies your query.

Comment: I understand about genotyping data. BIL1 was previously genotyped and analyzed for the QTLs of seed dormancy and use that linkage map. 
My another query is ....
You generated the genotype first for BIL1 (from seed dormancy QTL analysis experiment). This time you generated another population of BIL1 to evaluate the phenotype, and used the geneotype data from previous population, is it? 

Author Response

Thank you for your comment. We did not generate another population of BIL1. BIL1 used in this study is the same BIL in Ahamed et al. (2023).

BIL stands for Backcrossed recombinant Inbred Lines. BILs are generated by crossing two parents and then backcrossing the F1 plant with one of the original parents. The BC1F1 plants are self-pollinated for several generations to fix the genotypes. Therefore, each plant in the BIL population is a genetically pure line, with all genes in each line in a homozygous state.

In this study, we produced BC1F1s and utilized the single-seed descent method to purify the genotype (BC1F5), as stated in Line 77.

This allowed us to obtain sufficient seeds for each line, facilitating the evaluation of the same populations for various traits in different experiments. Therefore, the BIL1 population evaluated for heat stress in this paper is identical to the genotype evaluated for seed dormancy in Ahmed et al. (2023).

Round 4

Reviewer 1 Report

Comments and Suggestions for Authors

Thank you for your comment. We did not generate another population of BIL1. BIL1 used in this study is the same BIL in Ahamed et al. (2023).

BIL stands for Backcrossed recombinant Inbred Lines. BILs are generated by crossing two parents and then backcrossing the F1 plant with one of the original parents. The BC1F1 plants are self-pollinated for several generations to fix the genotypes. Therefore, each plant in the BIL population is a genetically pure line, with all genes in each line in a homozygous state.

In this study, we produced BC1F1s and utilized the single-seed descent method to purify the genotype (BC1F5), as stated in Line 77.

This allowed us to obtain sufficient seeds for each line, facilitating the evaluation of the same populations for various traits in different experiments. Therefore, the BIL1 population evaluated for heat stress in this paper is identical to the genotype evaluated for seed dormancy in Ahmed et al. (2023).

Comments: Thank you for the clarification. Please add a line to clarify this in the manuscript to the same population was used for genotyping and phenotyping for heat stress and seed dormancy. Otherwise, reader will confused.